# Cytochrome P450 Enzymes and Drug Metabolism in Humans

**DOI:** 10.3390/ijms222312808

**Published:** 2021-11-26

**Authors:** Mingzhe Zhao, Jingsong Ma, Mo Li, Yingtian Zhang, Bixuan Jiang, Xianglong Zhao, Cong Huai, Lu Shen, Na Zhang, Lin He, Shengying Qin

**Affiliations:** 1Bio-X Institutes, Key Laboratory for the Genetics of Developmental and Neuropsychiatric Disorders (Ministry of Education), Shanghai Jiao Tong University, Shanghai 200030, China; zhaomingzhe@sjtu.edu.cn (M.Z.); limo1169718691@sjtu.edu.cn (M.L.); zhangyingtian@sjtu.edu.cn (Y.Z.); bixuanjiang@sjtu.edu.cn (B.J.); zhaoxianglong2011@sjtu.edu.cn (X.Z.); huaic@sjtu.edu.cn (C.H.); mailer.shen@gmail.com (L.S.); zhangnazn@sjtu.edu.cn (N.Z.); helin@sjtu.edu.cn (L.H.); 2Institutes for Shanghai Pudong Decoding Life, Shanghai 200135, China; majingsong@westlake.edu.cn

**Keywords:** cytochrome P450, drug metabolism, genetic polymorphisms, protein structure

## Abstract

Human cytochrome P450 (CYP) enzymes, as membrane-bound hemoproteins, play important roles in the detoxification of drugs, cellular metabolism, and homeostasis. In humans, almost 80% of oxidative metabolism and approximately 50% of the overall elimination of common clinical drugs can be attributed to one or more of the various CYPs, from the CYP families 1–3. In addition to the basic metabolic effects for elimination, CYPs are also capable of affecting drug responses by influencing drug action, safety, bioavailability, and drug resistance through metabolism, in both metabolic organs and local sites of action. Structures of CYPs have recently provided new insights into both understanding the mechanisms of drug metabolism and exploiting CYPs as drug targets. Genetic polymorphisms and epigenetic changes in CYP genes and environmental factors may be responsible for interethnic and interindividual variations in the therapeutic efficacy of drugs. In this review, we summarize and highlight the structural knowledge about CYPs and the major CYPs in drug metabolism. Additionally, genetic and epigenetic factors, as well as several intrinsic and extrinsic factors that contribute to interindividual variation in drug response are also reviewed, to reveal the multifarious and important roles of CYP-mediated metabolism and elimination in drug therapy.

## 1. Introduction

Drug metabolism is the process of altering their molecules chemically after entering the body [1]. In general, the metabolism of drugs decreases their therapeutic effects [2]. The majority of drugs lipophilic centers are converted to hydrophilic centers during drug biotransformation, which can increase their water solubility, to allow elimination in urine or bile [3]. This is an important progress for drug metabolism, because the lipophilic nature of drugs can keep them staying for longer in the body, which may in turn lead to toxicity [4,5]. Drug metabolism can be divided into phase I and phase II reactions [6]. Figure 1 shows the known generalized pathways associated with drug metabolism catalyzed by cytochrome P450 (CYP) enzymes. Phase I reactions introduce reactive or polar groups (-OH, -COOH, -NH_2_, -SH, etc.) into drugs, including oxidation, reduction, and hydrolysis, where drugs cannot be excreted from bodies. The modified drugs are then conjugated to polar compounds in phase II reactions, which are catalyzed by a variety of transferase enzymes, such as uridine diphosphate (UDP)-glucuronosyltransferases, sulfotransferases, and glutathione S-transferases [7]. The conjugated drugs may be further processed, before being recognized by efflux transporters and pumped out of cells. However, the same metabolic process can also lead to the generation of reactive metabolites, which are toxic to the human body. This is termed the bioactivation of drugs, which depends specifically on important structural feature present in these compounds.

Drug metabolism is the metabolic breakdown of drugs through specialized enzymatic systems [8]. CYPs are involved in more than 90% of the reported enzymatic reactions [3]. Regarding drug metabolism, CYPs are the most well-known drug-metabolizing enzymes and are mainly expressed in the liver [9], but other organs are also involved: kidney, placenta, adrenal gland, gastrointestinal tract, and skin [10]. Among the 57 putatively functional human CYPs, the isoforms belonging to the CYP1, 2, and 3 families are mainly responsible for the metabolism of about 80% of clinical drugs [11]. CYP-mediated drug metabolism not only converts lipophilic products into hydrophilic products to facilitate elimination, but also plays a critical role in determining treatment outcomes, by influencing drug action, safety, bioavailability, and drug resistance, through the metabolism in both metabolic organs and local sites of action [12]. CYPs, as the most diverse catalysts known in biochemistry, contribute to interindividual variations in drug responses, resulting from genetic and epigenetic variants, as well as environmental factors, such as gender, age, nutriture, disease states, and pathophysiological factors [13]. In particular, CYPs can be inhibited or induced by concomitant drugs and circulating metabolites, which can influence treatment outcomes through drug–drug interaction (DDI), drug–gene interaction (DGI), and drug–drug–gene interaction (DDGI) [14].

It is worth emphasizing that CYPs are the most abundant and significant, as well as diverse, drug-metabolizing enzymes, and they play important roles in clinical drug metabolism [15]. In this review, we mainly concentrated on human CYPs; early research about CYPs necessarily involved animal models, but the intention was always to understand the human systems in the context of enzymes catalyzing the observed transformations. We covered the structures of CYPs, which have been discovered continuously, since the first was identified in the early 1980s. The wealth of new structural information has been particularly useful for giving a better understanding of CYP dynamics and how their active site adapts to substrates of diverse sizes and shapes. Of particular interest is the varying responses of individual patients to administered pharmaceuticals; thus, interindividual variations of drug metabolism resulting from genetic and epigenetic variants, as well as environmental factors, were systematically summarized. Lastly, we outline the clinical implications and therapeutic benefits of CYPs. With advances in molecular biology and biochemical technology, our knowledge of these critical metabolic process will eventually assist in the development of individualized pharmacotherapy, avoiding harmful adverse drug reactions or treatment failures.

## 2. Human CYPs

CYPs are the major enzymes involved in human drug metabolism (Figure 2). In looking at the fraction of drugs processed by enzymes, CYPs account for ~75%. The human genome encodes at least 57 CYPs, and these genes are organized into 18 families and 43 subfamilies (Table 1). CYPs play important roles in the maintenance of general human health, particularly as they relate to the metabolism of pharmaceuticals (Appendix A). Of salient interest for CYPs in drug metabolism is the varying response of individual patients to administered pharmaceuticals [16]. It is known that some individuals metabolize drugs relatively rapidly, while other individuals metabolize the same drugs relatively slowly [17]. The differences of metabolism may be associated with the expression of CYPs, particularly in the liver and intestines [18]. Some external factors, such as diet, prior exposure to other drugs, and tobacco and alcohol consumption have been suggested as influencing the expression and functional activity of CYPs that are closely related to endogenous substrates. 

A second realm of strong biomedical interest is the role of CYPs in the metabolism of antitumor agents. CYPs have been detected in tumors, as well as cancer cells and cell lines [19,20]. Most antitumor agents that exert antitumor efficacy in cancer cells have been observed to be metabolized by the CYP1, CYP2, and CYP3 family, such as flavonoids by CYP1b1, tamoxifen by CYP2D6, docetaxel and cyclophosphamide by CYP3A4/5, thalidomide by CYP2C9 and CYP2C19, and paclitaxel by CYP2C8 [21,22]. Thus, the expression of CYPs in tumor cells may play an important role in antitumor therapy. Of note, it was shown that the expression of CYPs in tumor cells was usually aberrant, compared with adjacent normal tissues [23]. The low expression and activity of CYPs, partly owing to distinctive metabolic reprogramming and living conditions, may reduce the activation of antitumor agents in tumor cells, whilst the overexpression of CYPs in tumor cells may rapidly devitalize tumor agent substrates, which may be associated with treatment resistance and cause subsequent tumor relapse and poor prognosis [24,25]. Accordingly, CYPs have been considered as targets and indicators for antitumor therapy because of their aberrant expression in tumor cells [26,27]. Several studies have emphasized the role of CYP1B1 in tumor progression and treatment resistance, recommending CYP1B1 as a novel oncological therapeutic target [28,29,30]. The development of several CYP1B1 inhibitors has been proposed to overcome treatment resistance in number of tumor cell lines and is regarded as the predominant therapeutic paradigm to treat malignancy [31]. In addition, several other CYPs have emerged as potential targets and indicators, such as CYP2J2 for breast cancer [32] and CYP2W1 for colon cancer [33]. Targeting CYPs in preclinical and clinical trials for chemoprevention and chemotherapy has become an effective way to improve antitumor treatment outcomes. 

## 3. Structures of CYPs

The CYPs are hemoproteins; embracing about 400–500 amino acids in their sequences and a single heme prosthetic group in the active site [34]. There now are 104 unique structures of CYPs that have been deposited in the Protein Data Bank (PDB), and this accumulating evidence suggests that the overall CYP folds are quite conservative. Members of the CYP family share about 40% sequence homology; with 55% sequence identity shared between subfamilies [35]. To date, nonheme proteins with CYPs folds have not been discovered, and a small handful of enzymes, including the CYP450nor [36], prostacyclin synthase [37], and allene oxide synthase [38], with CYPs folds do not catalyze traditional CYP chemistry. All CYPs involve a heme–iron center in the active site, tethered by a cysteine thiolate ligand localized in a characteristic FXXGXXXCXG element in their amino acid sequence. The shared tertiary structures usually include 12 common helices (A-L) and four common β-sheets. The structures of four CYPs are shown in Figure 3, although the overall folds of four CYPs are maintained, the precise position of structural elements varies substantially. Some key secondary structural elements are also highlighted in CYP101 in Figure 3. The closer to the heme, the more conserved is structure; especially helices I and L, which connect to the heme directly. The most conserved elements of CYPs center on heme–thiolate oxygen activation chemistry, such as the β-bulge segment housing the Cys ligand. Another highly conserved region involved in O_2_ activation is the portion of helix I near the heme. An outstanding structural characteristic of CYPs is their ability to adapt to substrates of various sizes and shapes. Most of our detailed understanding of CYP–substrate interaction derives from highly specific CYPs that bind to their respective substrates tightly. The size and shape of the various substrates for CYPs are fairly diverse. Substrates usually enter the active site near the connection between the F and G helices, which is the main entry point for substrates in many CYPs. The structural changes of regions including F and G helices in CYPs may be responsible for the requirement for substrate specificity [39]. CYP101 and cytochrome P450epoK represent the two extremes of substrate size and shape. Some of the most different regions when comparing these two enzymes are the F, G, and B’ helices. The B’ helix is rotated 90° in cytochrome P450epoK compared to CYP101. This reorientation opens the substrate-binding pockets, making room for specific regions of the substrates [40].

Due to technological advances in protein expression and purification, more structures of CYPs will be found. However, this field is now at the stage where structure discovery outpaces functional and biological studies. Some structures are now being determined before we know much about their functions. Structural information should be used to guide functional and biological studies, especially in the field of drug metabolism.

## 4. Characteristics of Major Drug Metabolizing CYPs

Among the 57 functional CYPs, the isoforms belonging to the CYP1, CYP2, and CYP3 families are responsible for the metabolism of around 80% of clinical drugs [11]. These include CYP1A2, CYP2A6, CYP2B6, CYP2C8, CYP2C9, CYP2C19, CYP2D6, CYP2E1, CYP3A4, and CYP3A5; with CYP3A4 and CYP2D6 contributing to over 50% of CYP-related drug metabolism (Figure 4). With broad coverage of oxidative metabolism across the CYP1, CYP2, and CYP3 families, each has unique characteristics. The CYP1A subfamily includes CYP1A1 and CYP1A2. CYP1A1 is distinct from other isoforms, which are mainly expressed in human liver and also have additional expressions in other tissues at varying levels [41], its major organ of expression is the lungs. Unlike most other drug metabolizing CYPs, CYP1A2 is exclusively expressed in the human liver. It was reported that CYP1A1 and CYP1A2 might play important roles in carcinogen bioactivation, particularly with aromatic and heterocyclic amines. Work with animal models has shown that CYP1A1 inducers can be cocarcinogens [42]. Thus, regulatory agencies have tended to look unfavorably on the induction of CYP1A1 by potential drugs in animal models. There is some epidemiological evidence that high CYP1A2 activity is associated with increased risk of colon cancer, although the effect was not seen in the absence of high *N*-acetyltransferase activity and high consumption of charbroiled meat [43]. CYP2 is the largest family of CYPs, and CYP2D6 and CYP2C9 from the CYP2 family are the highest contributors to drug metabolism. There is little or no overlap between the substrates catalyzed by each CYP2 isoform, which have very different active sites. CYP2D6 prefers basic molecules, whereas CYP2C9 prefers compounds with slightly acidic properties [44]. The clinical issues regarding CYP2D6 are considerable, due to the large variation in genetics in the population. An interesting issue regarding CYP2C9 involves the drug tienilic acid. The compound is a substrate and a mechanism-based inactivator of CYP2C9. Some patients treated with tienilic acid develop liver injury, while some patients treated with it also present with liver–kidney microsomal antibodies in their blood. All isoforms in the CYP2 family are predominantly express in the human liver, except for CYP2J2, which is reported to be primarily a cardiovascular CYP. CYP2J2 is associated with the etiology of several diseases, including hypoxia, cardiotoxicity, and coronary artery disease. The common inducers for most isoforms in the CYP2 family are Rifampicin and Artemisinin, but each isoform has well-accepted inhibitors, useful for selective in vitro studies. The drug metabolizing CYP3A subfamily plays an important role in both drug discovery and development. The CYP3A subfamily (specifically CYP3A4 and CYP3A5) is responsible for the metabolism of over 30% of drugs used today and is the most abundant CYP in the human body [45]. One strategy to improve the predictability in drug development is the use of transgenic ‘humanized’ mice expressing CYP3A4, which have been developed using different approaches [46,47]. Unlike the isoforms in the CYP2 family, CYP3A4 and CYP3A5 have an increased number of overlapping substrates. CYP3A4 covers a very diverse set of structures and has lipophilicity; it sometimes can accommodate two substrates at once and has been well characterized for broad substrate specificity.

## 5. Individual Variation of CYP-Mediated Drug Metabolism

The expression and activity of CYPs can vary considerably among individuals and ethnicities. Genetic variability in CYP genes has received great emphasis for explaining individual differences over the last two decades [22,48]. The polymorphisms of CYP genes are involved in multiple allelic variants, the frequencies of which vary among different populations [49,50]. More than 350 functionally polymorphic CYPs have been collected in the human CYP allele nomenclature committee home page (Date of access: 15 September 2021; http://www.pharmvar.org/ Version 5.1.3 lasted updated 6 November 2021). The highest amounts of allelic variants are described for *CYP2D6* (63 alleles), *CYP2B6* (28 alleles), and *CYP2A6* (22 alleles) [48]. CYP2D6, as the most common mutant isoform, is involved in the metabolic process of nearly 25% of clinical drugs, and its polymorphisms can affect the metabolic process of about 50% of these [51]. Accumulating evidence indicates that loss-of-function variants and gain-of-function variants are the two main types of genetic variation in CYP genes [52]. Loss-of-function variants, which frequently affect splicing and expression of CYP genes, may reduce elimination and enhance drug plasma concentrations [53], whilst gain-of-function variants, resulting from copy number variants with an increased number of functional gene copies, or promoter variants and amino acid variants with an increased substrate turnover of CYP genes, may enhance elimination and reduce drug concentrations [54].

There now are four types of phenotypical changes in CYPs that have been identified, including poor metabolizers (PM), intermediate metabolizers (IM), extensive metabolizers (EM), and ultra-rapid metabolizers (UM), which are attributed to drug response based on genetic variations in CYP genes [55]. PM usually suffer more adverse reactions at a normal dose of drug, due to being homozygous for either functionally variant alleles or due to a complete deletion of the gene causing reduced enzyme activity [56]. IM are heterozygous for specific variant alleles. EM have two functionally competent alleles [44]. UM with two or more active genes on the same allele often fail to respond to drugs at a normal dose [44]. Therefore, genetic polymorphisms in CYP genes may play important roles in the optimization of drug treatments with respect to efficacy and prediction of adverse reactions [48].

In addition to gene polymorphisms, epigenetic mechanisms, such as DNA methylation, which can regulate expression of CYP genes by targeting either the promoter region or upstream transcriptional factors, can also affect the variability of CYPs [49,57]. DNA methylation can influence the expression of some CYP genes, especially those involved in the metabolism of endogenous compounds [57,58]. It was reported that DNA methylation in the promoter of genes switched off CYP gene expression, by rejecting the binding of some transcription factors to their DNA binding sites [59]. Some functional methylation sites have been found in CYP genes, including CYP1A1, CYP1B1, CYP2W1, CYP2C19, and CYP2D6 [60,61].

The noncoding RNAs, such as miRNAs, can also influence the interindividual variability of CYP expression involved in various cellular processes like proliferation, morphogenesis, apoptosis, and differentiation [62]. It was suggested that the probability of potential sites for miRNA regulation of CYPs depends on the size of the 3′-UTR region; the extent of regulation being directly proportional to the length of the region [63,64]. In addition, genetic variants in the mRNA target binding sites or in the miRNA precursor may also lead to variable expression of CYP genes.

The interindividual variability of CYP-mediated drug metabolism can also be affected by environmental factors, i.e., intrinsic factors (age and disease states) and extrinsic factors (nutrition and smoking), as well as comedication (induction and inhibition), which can be important for predicting how an individual will respond to a drug [48]. Central nervous system (CNS)-acting drugs often target the human brain in the therapy of CNS disorders, such as schizophrenia, major depressive disorder, and anxiety disorder etc. [65]. Most CNS-acting drugs are metabolized by CYPs, especially the CYP2 family [66]. Some CYPs in the CYP2 family usually change more with age [66]. It was shown that CYP2D6 often remains at a low level at birth and increases gradually with age until reaching the highest levels at 65 years old [67]. The CYP2D6 in liver usually increases quickly to adult levels after birth and keeps constant with age [68]. The pharmacologic effects of CNS-acting drugs depend on their availability and the levels reached in the human brain; the expression of CYPs may influence the cerebral levels of drugs, causing different therapeutic outcomes [69]. In addition to age, disease states, as another common intrinsic factors, can also influence CYP expression, which may have a negative effect on the metabolic capacity of drugs [70]. As mentioned in Section 2, antitumor drug-metabolizing CYPs may be aberrantly expressed in tumor cells, because of their involvement in tumor physiology and pathology, such as the overexpression of both CYP1B1 in breast cancer cells and CYP2A6 in liver and lung cancers [71,72,73,74]; while, the expression of some CYPs involved in the development of liver ischemia, reperfusion, and sepsis are decreased [75]. Infection and inflammation states can also contribute to interindividual variability of drug response, by regulating the expression and activities of drug-metabolizing CYPs [76].

As reported, smoking and nutrition are associated with the activity variation of CYPs [77,78]. It was shown that smokers had higher CYP2D6, CYP2E1, and CYP2B6 levels compared with nonsmokers [69,79]. In addition to smoking, some dietary chemicals may regulate the catalytic activity of CYPs. For example, an increase of unsaturated fatty acids in food can enhance the expression of CYPs in the liver [80], and lacking protein, vitamin C, calcium, or magnesium in food may reduce the activity of CYPs in the process of metabolizing some drugs [81,82,83]. CYP3A can be induced by some brassicaceous vegetable, such as turnips and spinach, causing the enhancement of the first-pass effect of phenacetin [84]. On the contrary, CYP3A can be inhibited by grapefruit juice, which is rich in bioflavonoids and naringin, leading to a decrease in the first-pass effect of felodipine, nifedipine, midazolam, and cyclosporine [85].

The CYPs usually include both active sites and allosteric sites, where drug molecules can selectively bind as inducers or inhibitors [86]. It was reported that CYP induction or inhibition is a major mechanism underlying DDI [87,88]. The specific process of this mechanism is complicated, because multiple occupancies and multistep bindings make CYPs susceptible to being induced or inhibited [89]. Metabolite intermediates can also exert induction or inhibition on CYPs and impact the metabolism of drugs catalyzed by the same CYPs [90]. In addition, genetic variants that affect the expression and activity of CYPs may have an impact on DDI through DDGI, with a cumulative effect on both DDI and DGI [91,92].

CYP induction is a process that is relatively common among the CYPs involved in the oxidation of xenobiotic chemicals (Appendix A) [93]. It is mostly transcriptional regulation, and mainly resulting from epigenetic regulation, although non-transcriptional mechanisms, such as enzyme stabilization, stabilization of mRNA, or inhibition of protein degradation, have also been reported [94]. Several major systems are known to be involved in the induction of CYPs. The aryl hydrocarbon receptor (AhR) system involves the AhR and AhR nuclear tanaporter proteins, regulating CYP1A1, CYP1A2, CYP1B1, and CYP2S1. In addition, three distinct ‘orphan receptors’, which belong to the nuclear receptors, have also been identified. These include nuclear pregame X receptor (PXR), which activates CYP3A genes in response to varying chemicals, including synthetic and natural steroids [95]; the constitutive androstane receptor (CAR), which mediates the induction of CYP2B genes by phenobarbital [96]; and the peroxisome proliferator-activated receptor (PPAR), which mediates induction of the fatty acid hydroxylases of the CYP4A family [97]. CAR and PXR are the major nuclear receptors related to CYP induction are activated by clinical drugs [98]. After the direct activation of inducers, these nuclear receptors will enter the nucleus to bind with the response elements in DNA, with the synergy of recruited coactivators affecting the chromatin structure, and finally contributing to the augmentation of the target gene transcription [98]. Moreover, CYPs can be activated indirectly by inducers such as phenobarbital, which can activate the CAR by inhibiting the epidermal growth factor receptor [99].

CYP inhibition is generally more common than induction (Appendix A). It is considered as a principal mechanism for metabolism-based DDI and usually involves competition with another drug for the same CYP binding site [88,100]. CYP inhibition can damage the biotransformation or clearance of all clinically used drugs, causing higher plasma levels of drug, and further affecting therapeutic responses and increasing the chances of adverse drug reactions [88,100]. As indicated, inhibition of CYPs can be categorized into two basic types: reversible inhibition and irreversible inhibition [101]. Reversible inhibition includes competitive inhibition, uncompetitive inhibition, and noncompetitive inhibition [94]. Competitive inhibition is a form of enzyme control in which an inhibitor molecule, very similar in structure to the normal substrate of an enzyme, becomes reversibly bound to the active site, thus reducing the quantity of enzyme available [102] (Figure 5a). Competitive inhibition occurs when two drugs compete for the same CYP, irrespective of whether they are substrates for that CYP [103]. Some inhibitors of CYP3A4 that act by this mechanism of inhibition include the azole antifungal agent ketoconazole. Uncompetitive inhibition is an inhibitory effect on a metabolic function, such as that of CYPs, and not based on competition for the binding site of the naturally occurring substrates, but on a different effect on the molecule, the function of which is being inhibited [104] (Figure 5b). Uncompetitive inhibition binds only to the enzyme–substrate complex. In fact, competitive inhibition is more common, while uncompetitive inhibition is relatively rare [105]. Noncompetitive inhibition is a type of enzyme inhibition in which the inhibiting compound does not compete with the natural substrate for the active site on the enzyme, but inhibits the reaction by combining with the enzyme–substrate complex, as well as with the free enzyme [106] (Figure 5c). Many drugs are noncompetitive inhibitors of CYPs, such as omeprazole, lansoprazole, and cimetidine. The key difference between competitive inhibition and noncompetitive inhibition is that in competitive inhibition, the binding of an inhibitor prevents the binding of the target molecule with the active site of the enzyme whereas, in noncompetitive inhibition, an inhibitor reduces the activity of an enzyme. Irreversible inhibition is the second type of CYP inhibition, in which inhibitor binds with the enzyme by a strong covalent bond and inhibits the enzyme activity. Irreversible inhibition is usually caused by metabolite intermediates that can be restored with a new synthesis, which makes the irreversible inhibition more severe [107]. There are three types of irreversible inhibitors, including group-specific reagents, substrate analogues, and suicide inhibitors. Both reversible inhibition and irreversible inhibition can change the catalytic activity of CYPs. The key difference between reversible and irreversible inhibition is that it is possible to reverse reversible inhibition, while it is not possible to reverse irreversible inhibition. CYP inhibition has the same effect as a genetic deficiency (attenuation of drug metabolism, leading to increased pharmacological response), but this can be even more problematic, because of temporal changes. For instance, some clinical drugs can produce a delayed response, and the pharmacokinetics of a substrate may vary with time. Consequently, CYP inhibition is an important practical matter for drug discovery, development, and in clinical practice.

Varying drug concentrations in plasma have been reported in DDIs when a given drug induced or inhibited one CYP metabolism pathway, and the genetic variation altered the other pathway [108]. This overlapping between DDI and DGI is referred to as DDGI, which can be considered as a combined effect of a genetic variant with the perpetrator drug on the multiple drug metabolic pathways [92,109]. The effect of DDGI should be considered when a perpetrator drug is prescribed for a patient stabilized with a victim drug or when a victim drug is administered to a patient who has been prescribed a perpetrator drug [110]. However, there have been limited published studies and insufficient research on the prevalence and evaluation models of clinical DDGI. 

## 6. Clinical Implications and Therapeutic Benefits

The induction and inhibition of CYPs, which can mediate DDI and the bioactivation of xenobiotics is profound and clinically important [111]. CYP induction of various active parent drugs can result in increasing the metabolism and elimination of drugs; thereby, diminishing their therapeutic effect. CYP inhibition can result in either drug accumulation or decreased drug metabolism, leading to possible clinical toxicity or enhancement of pharmacological effects [112].

Variants of CYP genes have major impacts on individual variability in drug response and therapeutic outcomes [113]. Genotyping and phenotyping tests for CYPs are increasingly being conducted in clinical practice to identify patients who are at risk of drug inefficacy or toxicity and to implement individual treatments. The ultra-rapid metabolizer phenotype is associated with poor therapeutic efficacy of drugs, and the poor metabolizer phenotype is responsible for the toxicity of drugs [114,115]. Pharmacogenetic-based dosing for drugs could be very useful if robust studies suggested the benefit of pre-emptive genotyping was associated with better outcomes. For example, dose reductions are recommended in CYP2C19 poor metabolizers, to avoid the risk of adverse effects. Samer and colleagues have published a consensus guideline for dose recommendation, based on CYP pharmacogenomics testing [113]. The Clinical Pharmacogenetics Implementation Consortium (CPIC) have published several guidelines that enable the translation of genetic test results into actionable prescription decisions for drugs [116,117,118].

## 7. Conclusions and Future Perspectives

Although research on CYPs in drug metabolism has been conducted for several decades, many questions and challenges still exist. With technological advances in protein expression and purification, as well as the increasing genome databases, the crystal structures of CYPs are continually being solved. Artificial-intelligence (AI) approaches are being applied to the prediction of the 3D structures of proteins. Although these approaches are not yet accurate enough to be widely used in drug design, they are starting to be useful to crack proteins’ functions. Genetic polymorphisms that contribute to the variation of CYP phenotypes among humans can partly explain the interindividual differences in drug response. The potential use of CYP polymorphisms in developing personalized medicine is one of the most important challenges ahead. Epigenetic mechanisms, such as DNA methylation and miRNA, play important roles in the regulation of CYP gene expression and function. There is scope for further studies to explore the influence of epigenetic regulation on interethnic and interindividual variations in drug responses. Physiologically based pharmacokinetic models have been proposed as excellent tools to explore the potential DDGIs of drugs. In addition, pharmacogenetics of DDIs and DDGIs should be given full consideration in the future.

## Figures and Tables

**Figure 1 ijms-22-12808-f001:**
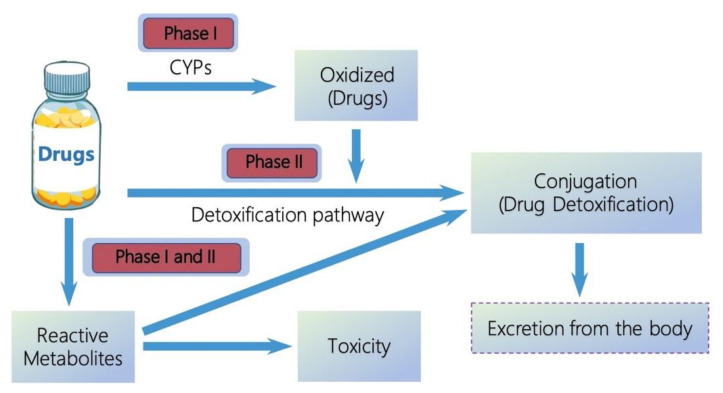
General pathways of drug metabolism.

**Figure 2 ijms-22-12808-f002:**
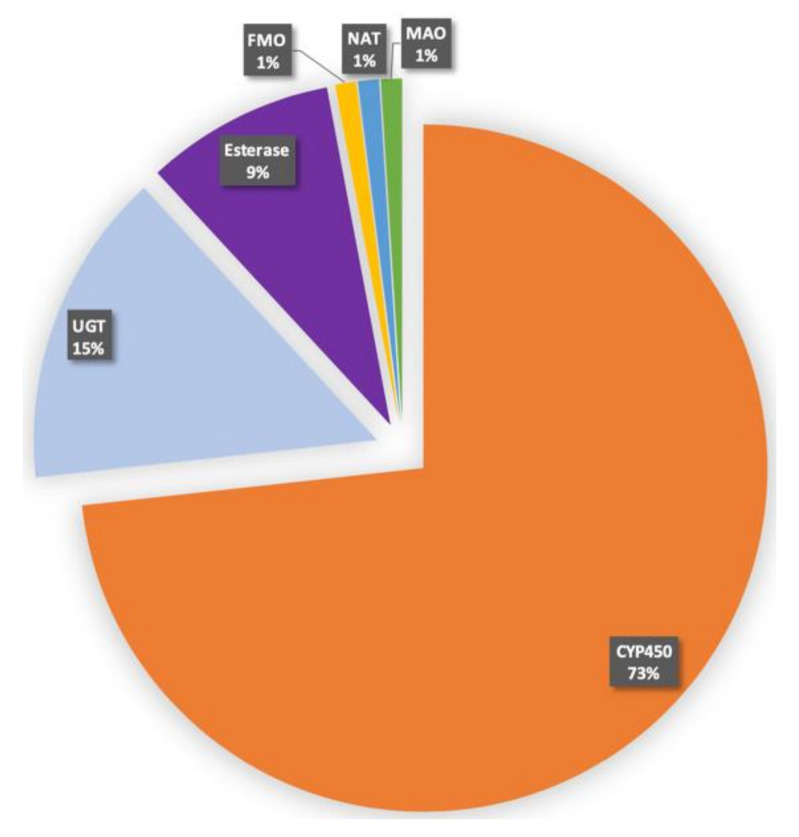
Contribution of different enzymes to drug metabolism. UGT, UDG glucuronosyl transferase; FMO, flavin-containing monooxygenase; NAT, N-acetyltransferase; MAO, monoamine oxidase.

**Figure 3 ijms-22-12808-f003:**
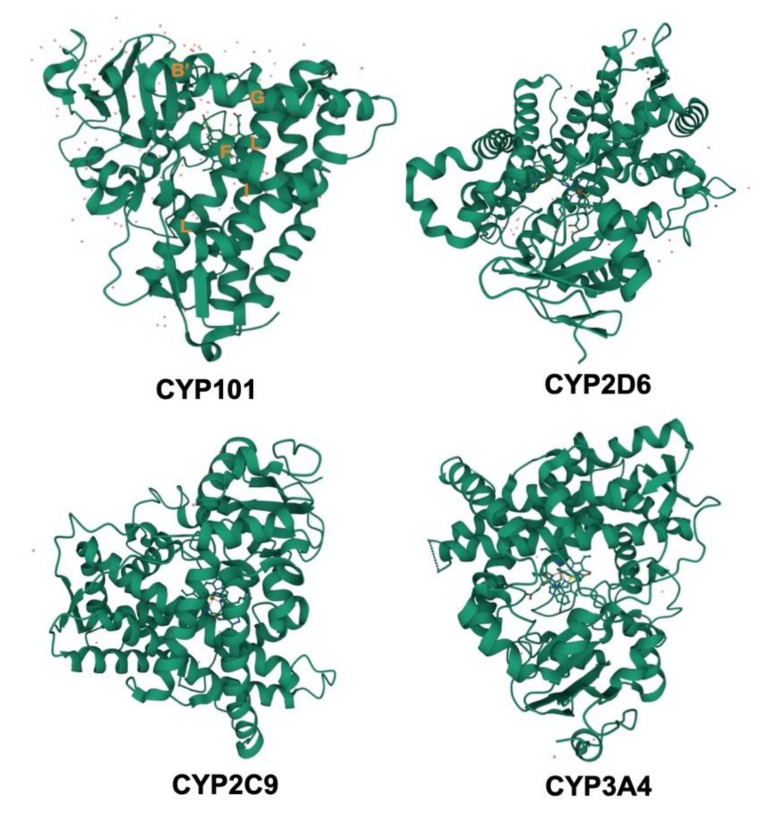
A representative example of known CYP structures, illustrating the common three-dimensional fold.

**Figure 4 ijms-22-12808-f004:**
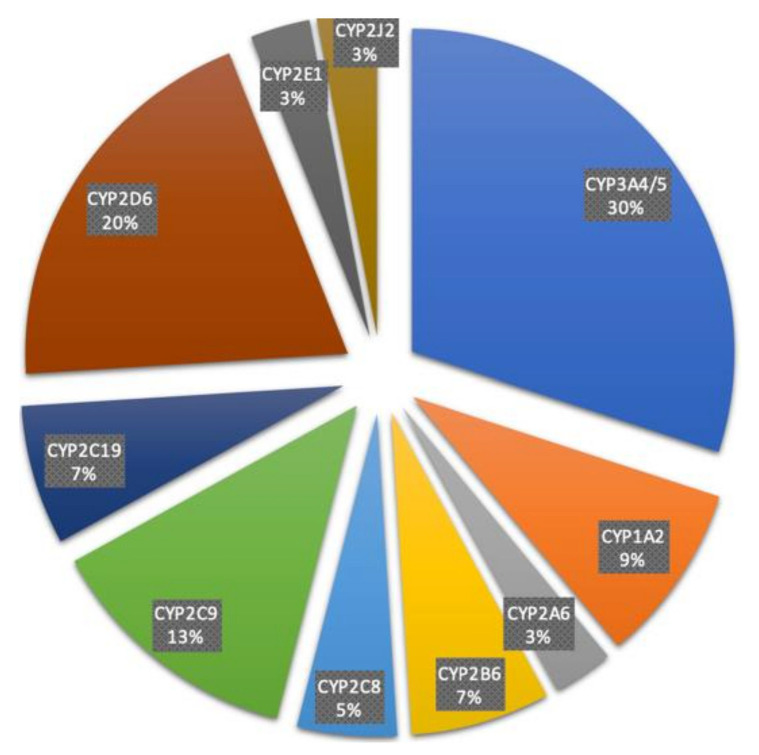
Fraction of specific CYP isoforms contribution to 248 drug metabolisms.

**Figure 5 ijms-22-12808-f005:**
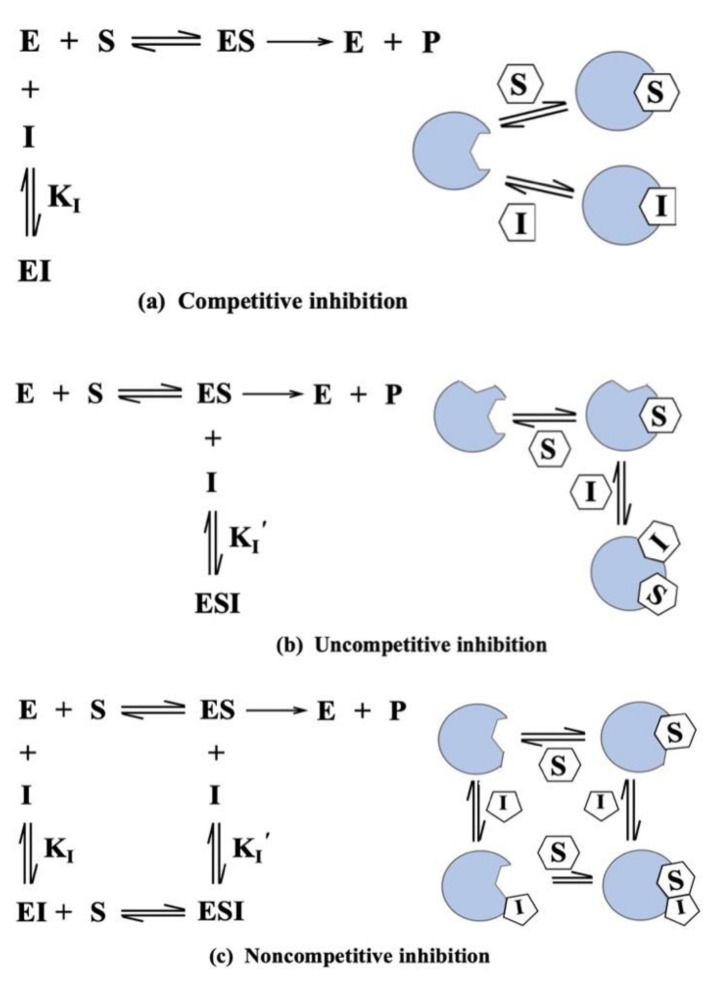
Types of reversible inhibition.

**Table 1 ijms-22-12808-t001:** Human CYPs diversity and functions.

CYP Family	Primary Functions	Subfamilies	Genes
1	drug metabolism	3	3
2	drug/steroid metabolism	13	16
3	drug metabolism	1	4
4	arachidonic acid/ fatty acid metabolism	5	12
5	thromboxane synthase	1	1
7	steroid 7α-hydroxylase	2	2
8	bile acid biosynthesis; prostacyclin synthase	2	2
11	steroid biosynthesis	2	3
17	steroid 7α-hydroxylase	1	1
19	aromatase	1	1
20	function not determined	1	1
21	steroid biosynthesis	1	1
24	vitamin D deactivation	1	1
26	retinoic acid hydroxylase	3	3
27	bile acid biosynthesis; vitamin D3 activation	3	3
39	function not determined	1	1
46	cholesterol 24-hydroxylase	1	1
51	lanosterol 14α-demethylase	1	1

## Data Availability

Not applicable.

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
