# Peer review of "Cytochrome P450 Enzymes and Drug Metabolism in Humans"

_ijms, 2021, doi:10.3390/ijms222312808_

Round 1
Reviewer 1 Report
The Authors of the manuscript summarize the basic knowledge about cytochromes P450 (CYPs) and major CYPs in drug metabolism in humans. Overall, it is an interesting work, based on current publications on CYPs. However, as a review article, it is very general, covers many known issues with the CYPs, but lacks some specific main topic, a new look at the role of CYPs in drug metabolism. In addition, the text sometimes gives too short description of what is shown in the figures. Perhaps it would be useful to consider the dynamics of CYPs in more detail and how their active site adapts to substrates of different sizes and shapes (the context of the substrate specificity of the enzymes)? Then the title should also suggest a new approach to the topic of CYPs. Therefore, in my opinion, the manuscript could only be published after major revision.
The following minor changes are suggested:
- Line 38: Mention also other, than glutathione S-transferases, phase II drug-metabolizing enzymes
- Line 42 (Figure 1 caption): Wouldn’t it be better: ‘General pathways…’?
- Please note that lines 43-59 are repeated on lines 60-76.
- Line 104: What do you mean by writing ‘endogenous CYPs’? CYPs are physiological enzymes of the organism, while their substrates can be considered exo- and endogenous.
- Line 108 (Table 1 caption): Wouldn’t it be better: ‘…and functions’?
- Please check the reference numbers on page 4 and ensure they are totally related. It looks like something is wrong, because references from [19] to [33] are missing in the text.
- Line 111: Please correct the typo in the word 'dingle'.
- Line 138 (chapter title): Write ‘Major’ in lowercase.
- Line 173: When referring to the website, you should enter the date of access.
- Line 263: Please write 'nuclear pregame X receptor'.
- Line 340: No space in ‘table1’.
- Supplementary: No space in ‘table1’. Do you need capital letters ('Major', 'Humans') in the table title? It is advisable to provide some references here.
- If it is possible the quality of Figures 2 and 5 should be improved.
Author Response
Dear Reviewer:
Thank you for your comments concerning our manuscript, “Cytochrome P450 enzymes and drug metabolism in humans”. These comments were invaluable for improving our paper and also provided ideas for future work. We have studied the comments carefully and have made corrections we hope will meet your approvals. Revisions to the manuscript are in colored text (Red) and detailed point-by-point responses to your comments are appended below.
The Authors of the manuscript summarize the basic knowledge about cytochromes P450 (CYPs) and major CYPs in drug metabolism in humans. Overall, it is an interesting work, based on current publications on CYPs. However, as a review article, it is very general, covers many known issues with the CYPs, but lacks some specific main topic, a new look at the role of CYPs in drug metabolism. In addition, the text sometimes gives too short description of what is shown in the figures. Perhaps it would be useful to consider the dynamics of CYPs in more detail and how their active site adapts to substrates of different sizes and shapes (the context of the substrate specificity of the enzymes)? Then the title should also suggest a new approach to the topic of CYPs. Therefore, in my opinion, the manuscript could only be published after major revision.
Response: Thank you for your comments. We very agree with your point that this review is very general and just covered some known issues about CYPs. It lacks some specific main topic in some ways and a new look at the role of CYPs in drug metabolism. We consider this review as an overview of association between CYPs and drug metabolism in human. We began this review with the human CYPs. We outlined two hot topics about CYPs in drug metabolism: varying responses of individual patients to administered pharmaceuticals and the role of CYPs in the antitumor agents. Secondly, we introduced the structure of CYPs because the structural information can be used to guide functional and biological studies in the field of drug metabolism. With technological advance in protein structure predicting, artificial-intelligence (AI) approaches are being applied to the prediction of the 3D structures of protein, such as the AlphaFold. The prediction of structure with AI technology is going to be another hot topic in the field of drug metabolism. Then we focused on the CYP1, CYP2, and CYP3 family of CYPs, which are mainly responsible for the metabolism of around 80% of clinical drugs. Lastly, we summarized and highlighted the individual variation of CYP-mediated drug metabolism. The genetic polymorphisms and epigenetic changes in CYP genes, as well as environmental factors were illustrated to explain the potential responsibility for individual variation of CYP-mediated drug metabolism. Particularly, we emphasized the drug-drug interaction, drug-gene interaction, and drug-drug-gene interaction in the inhibition and induction of CYPs. We will continue to focus on the field of CYPs in the drug metabolism and keep updating the relative review about the role of CYPs in the drug metabolism with more specific topics and new perspectives.
We have extended some description of what is shown in the figures.
(Perhaps it would be useful to consider the dynamics of CYPs in more detail and how their active site adapts to substrates of different sizes and shapes. (The context of the substrate specificity of the enzymes)?) It will be very significant if we provide more details about the dynamics of CYPs and the substrate specificity of CYPs. It should cover each of the 57 human CYP genes/genes products. Points to be covered with each CYP, when possible, include sites of expression and relative abundance, regulation, genetic variation, substrates and reactions, structure, inducers, inhibitors, and clinical issues. Given the main intention of this review, we didn’t conduct a comprehensive literature searching to cover all of this information. However, it will be more useful to build a database that collects all these characteristics of 57 human CYPs. Thanks again for your valuable advice that is very instructive for our study in the future.
Minor changes.
Mention also other, than glutathione S-transferases, phase II drug-metabolizing enzymes.
Response: Thank you for your comments. We have revised them in the manuscript.
Line 42 (Figure 1 caption): Wouldn’t it be better: “General pathways......”
Response: Thank you for your comments. We have changed it in the Figure 1 caption.
Please note that lines 43-59 are repeat on lines 60-76.
Response: Thank you for your reminding. We have revised it in the manuscript.
Line 104: What do you mean by writing “endogenous CYPs”? CYPs are physiological enzymes of the organism, while their substrates can be considered exo-and endogenous.
Response: Thank you for your comments. We are sorry that we didn’t make “endogenous CYPs” clear. We want to express the meaning that some external factors can influence the expression and functional activity of CYPs closely related with endogenous substrates. We have rewritten this sentence in the manuscript.
Line 108 (Table 1 caption): wouldn’t it be better: “...... and functions”?
Response: Thank you for your comments. We have changed it in the Table 1 caption.
Please check the reference numbers on page 4 and ensure they are totally related. It looks like something is wrong, because references from [19] to [33] are missing in the text.
Response: Thank you for your comments. I am so sorry that the second paragraph of Section 2 was replaced by the position of Figure 2 in moving the manuscript into template of IJMS. We have replenished the missing second paragraph with red text in the Section 2.
Line 111: Please correct the typo in the word “dingle”.
Response: Thank you for your reminding. We have corrected this word in the manuscript.
Line 138 (Chapter title): Write “Major” in lowercase.
Response: Thank you for your reminding. We have corrected it in the Chapter title.
Line 173: When referring to the website, you should enter the date of access.
Response: Thank you for your comments. We have entered the date of access for the website.
Line 263: Please write “nuclear pregame X receptor”.
Response: Thank you for your comments. We have revised it in the manuscript.
Line 340: No space in “table 1”.
Response: Thank you for your reminding. We have corrected it in the manuscript.
Supplementary: No space in “table1”. Do you need capital (Major, Human) in the table title? it is advisable to provide some references here.
Response: Thank you for your comments. We have corrected the title in the supplementary material and provided some references for every CYP.
If it is possible the quality of Figure 2 and 5 should be improved.
Response: Thank you for your comments. We have tried our best to improve them.
Reviewer 2 Report
Manuscript ID: ijms-1450839
Title: Cytochrome P450 enzymes and drug metabolism in humans
The manuscript described human cytochrome P450 (CYP) enzymes including the enzyme structures, some major enzyme isoforms, the individual variability caused by CYPs isoforms, and its related clinical implications.
- There were some front, wording and redundancy issues e.g. line173-177, CYP (line 24) and the paragraph (line 43 -59) has been repeated in the next paragraph (line 60-76).
- The efficacy of the drug and other biotransformation enzymes information was not probably showed in Figure 1
- A lot of details of the Section 4 (line 139-164) and Section 5 (line 168-186) were missed in Table 1.
- Why these four enzymes in Figure 3 have been selected to display? Especially, CYP101 was not highlighted with any other interests in the paper. The structure information in Figure 4 could be combined with Figure 3 (CYP101).
- Why the structure of the enzymes could not be studied before any further function studies (line 133-137)? Or what the author was emphasizing for this conclusion (line 133-137)?
- More examples were suggested for the majority of the section 4 and section 5 statements e.g. line 150-154, how was the functional contribution between CYP2C9 and CYP2C19.
- What is the logical linkage between Section 2 and paragraph (line 228-234)?
- Why noncompetitive inhibition has been mentioned in Section 5 (line 281-293)?
Author Response
Dear Reviewer:
Thank you for your comments concerning our manuscript, “Cytochrome P450 enzymes and drug metabolism in humans”. These comments were invaluable for improving our paper and also provided ideas for future work. We have studied the comments carefully and have made corrections we hope will meet your approvals. Revisions to the manuscript are in colored text (Red) and detailed point-by-point responses to your comments are appended below.
There are some front, wording and redundancy issues e.g., line 173-177, CYP (line 24) and the paragraph (line 43-59) has been repeated in the next paragraph (line 60-76).
Response: Thank you for your comments. We have revised them in the manuscript.
The efficacy of the drug and other biotransformation enzymes information was not probably showed in Figure 1.
Response: Thank you for your comments. Indeed, figure 1 just displayed a general pathway of drug metabolism catalyzed by the CYPs. We wanted to illustrate that drug metabolism is the biochemical modification of one chemical form to another, occurring usually through specialized enzymatic system, such as CYPs. In order to focus on topic of this review, we did not consider other enzymes. The drug metabolism will influence efficacy of drug, which has been discussed in other section of this review. As already reported, the mechanism of how drug metabolism influences the efficacy of drug is complicated thus we did not show it in this figure. It is a limitation of this flowchart without other enzymes information and the relationship between CYPs and efficacy of drugs. Thanks again for your valuable advice.
A lot of details of the Section 4 (line 139-164) and Section 5 (line 168-186) were missed in Table 1.
Response: Thank you for your comments. We used Table 1 to roughly display the overall characteristics of all human CYPs that is coded by 57 genes and organized into 18 families and 43 subfamilies. We wanted to specifically introduce the characteristics of three major drug metabolizing CYPs (CYP1, CYP2, and CYP3 family) which are responsible for the metabolism of around 80% of clinical drugs in Section 4. Although more than 350 functionally polymorphic CYPs have been reported, there are still a lot of unknown variants in CYPs waiting for exploring. As reported, the highest amounts of allelic variants are located in CYP2D6, CYP2B6, and CYP2A6. We could not collect enough information about variants in all CYPs thus we did not display them in Table 1. We just illustrated some individual variants with high frequencies that are associated with CYP-mediated drug metabolism in Section 5.
Why these four enzymes in Figure 3 have been selected to display? Especially, CYP101 was not highlighted with any other interests in the paper. The structure information in Figure 4 could be combined with Figure 3 (CYP101).
Response: Thank you for your comments. There now are more than hundred structures of CYP having been resolved. We selected four representative CYPs to display the common three-dimensional fold. CYP101 was the first CYP structure solved in the early 1980s and its structure is relatively familiar for researchers. As for the CYP2D6, CYP2C9, and CYP3A4, they are the top three CYPs contributing to drug metabolism as showed in Figure 5. Thus, their structural information had been particularly useful in drug discovery. We have combined the structural information in Figure 4 with Figure 3, which makes the figure more rational, thanks for your advice.
Why the structure of the enzymes could not be studied before any further function studies (line 133-137)? Or what the author was emphasizing for this conclusion?
Response: Thank you for your comments. The large increase in CYP crystal structures is due in large part to technological advance in protein expression and purification. The increasing genome databases make it relatively easy to discover new CYPs. Such advances are major contributors to the over-expanding number of structures deposited in the protein database. Many structures are being solved before one knows much about function. We thus must start using structural information to guide functional and biological studies. Thai could be particularly important with orphan CYPs that will continue to increase in number as more and more CYPs are discovered in new and interesting places.
More examples were suggested for the majority of the Section 4 and Section 5 statement e.g., line 150-154, how was the functional contribution between CYP2C9 and CYP2C19.
Response: Thank you for your comments. We have revised them in the manuscript.
What is the logical linkage between Section 2 and paragraph (line 228-234)?
Response: Thank you for your comments. I am so sorry that the second paragraph of Section 2 was replaced by the position of Figure 2 in moving the manuscript into template of IJMS. We have replenished the missing second paragraph with red text in the Section 2.
Why noncompetitive inhibition has been mentioned in Section 5 (line 281-293)?
Response: Thank you for your comments. Because CYP inhibition can be categorized into reversible inhibition and irreversible inhibition. Reversible inhibition includes competitive inhibition and uncompetitive inhibition. We wanted to introduced every category of CYP inhibition including uncompetitive inhibition.
Round 2
Reviewer 1 Report
The Authors responded in detail to all comments. The changes made the paper more interesting and readable. However, in some parts, it still needs to be clarified. For example, in Section 3, there are no examples of how the CYP active site adapts to substrates of different sizes and shapes, which is mentioned in lines 67 – 69. I think it is worth making a general discussion in the context of the presented CYPs, providing examples. The text still gives too short description of what is shown in the figures, e.g., lines 33 – 41 do not provide information on the toxicity caused by reactive metabolites presented in Figure 1.
Minor comments:
- Figure 1: Note that reactive metabolites can also undergo detoxification, which should be included in your scheme.
- Lines 91 – 92: Shouldn’t it be: ‘…in the metabolism of antitumor agents’?
- Lines 91 – 111: Some examples of antitumor drugs metabolized by CYPs would be indicated here.
- Lines 306 – 308: It is worth mentioning here the mechanism-based inhibition (as irreversible inhibition type) that CYPs undergo.
In my opinion, the manuscript could be published after minor revision.
Author Response
Dear Reviewer:
Thank you for your comments concerning our manuscript, “Cytochrome P450 enzymes and drug metabolism in humans”. These comments were invaluable for improving our paper and also provided ideas for future work. We have studied the comments carefully and have made corrections we hope will meet your approvals. Revisions to the manuscript are in colored text (Red) and detailed point-by-point responses to your comments are appended below.
The Authors responded in detail to all comments. The changes made the paper more interesting and readable. However, in some parts, it still needs to be clarified. For example, in Section 3, there are no examples of how the CYP active siteadapts to substrates of different sizes and shapes, which is mentioned in line 67-69. I think it is worth making a general discussion in the context of the present CYPs, providing examples. The text still gives too short description of what is shown in the figures, e.g., lines33-41 do not provide information on the toxicity caused by reactive metabolites presented in Figure 1.
Response: Thank you for your comments. We have discussed how the CYP active site adapts to substrates of different sizes and shape and provided an example to illustrate the possibility of the discussion. We also have provided information about the toxicity caused by reactive metabolites, which makes our description of Figure 1 more comprehensive. Thanks again for your valuable advice.
Minor comments.
Figure 1: Note that reactive metabolites can also undergo detoxification, which should be included in your scheme.
Response: Thank you for your comments. We have modified the Figure 1 in the manuscript.
Line 91-92: Shouldn’t it be: “.... in the metabolism of antitumor agents.”
Response: Thank you for your comments. Your advice makes our expression more accurate and we have revised it in the manuscript.
Lines 91-111: Some examples of antitumor drugs metabolized by CYPs would be indicated here.
Response: Thank you for your comments. We have provided some examples of antitumor drugs metabolized by CYPs in the manuscript.
Line 306-308: It is worth mentioning here the mechanism-based inhibition (as irreversible inhibition type) that CYPs undergo.
Response: Thank you for your comments. We have revised it in the manuscript.
Reviewer 2 Report
Manuscript ID: ijms-1450839
Title: Cytochrome P450 enzymes and drug metabolism in humans
The manuscript described human cytochrome P450 (CYP) enzymes including the enzyme structures, some major enzyme isoforms, the individual variability caused by CYPs isoforms, and its related clinical implications.
- The structure highlights of CYP101 are not easy to read (Figure 1), if possible, please change the front of the marker letters.
- Why noncompetitive inhibition has not been mentioned in Section 5 (line 301-304)? Please illustrate three types of reversible inhibition: competitive inhibition, uncompetitive inhibition, and noncompetitive inhibition with concept, distinguish and favorable examples. If possible, the following equations of different type inhibitions would be preferred.
Author Response
Dear Reviewer:
Thank you for your comments concerning our manuscript, “Cytochrome P450 enzymes and drug metabolism in humans”. These comments were invaluable for improving our paper and also provided ideas for future work. We have studied the comments carefully and have made corrections we hope will meet your approvals. Revisions to the manuscript are in colored text (Red) and detailed point-by-point responses to your comments are appended below.
The structure highlights of CYP101 are not easy to read, if possible, please change the front of the marker letters.
Response: Thank you for your comments. We have modified it in the Figure 3.
Why noncompetitive inhibition has not been mentioned in Section 5? Please illustrate three types of reversible inhibition: competitive inhibition, uncompetitive inhibition, and noncompetitive inhibition with concept, distinguish and favorable examples. If possible, the following equations of different type inhibitions would be preferred.
Response: Thank you for your comments. We have revised them in the manuscript and provided a figure to display the equations of different reversible inhibitions.
Round 3
Reviewer 2 Report
The manuscript needs some English language check such as “……the reaction by reaction by combining……” (line 332).
Author Response
Dear reviewer,
Thank you for your comments concerning our manuscript, “Cytochrome P450 enzymes and drug metabolism in humans”. These comments were invaluable for improving our paper and also provided ideas for future work. We have studied the comments carefully and have made corrections we hope will meet your approvals.
The manuscript needs some English language check such as “........ the reaction by reaction by combing.......” (Line332)
Response: Thank you for your comments. We have revised it in the manuscript and checked the whole manuscript again to avoid any errors.